# Design of the Depth Controller for a Floating Ocean Seismograph

**Haocai Huang** [1,2]![ID], **Chenyun Zhang** [1], **Weiwei Ding** [3], **Xinke Zhu** [3], **Guiqing Sun** [1] and **Hangzhou Wang** [1,*]![ID]

[1]   Ocean College, Zhejiang University, Zhoushan 316021, China; hchuang@zju.edu.cn (H.H.); cyzhang@zju.edu.cn (C.Z.); sgq@zju.edu.cn (G.S.)
[2]   Laboratory for Marine Geology, Qingdao National Laboratory for Marine Science and Technology, Qingdao 266061, China
[3]   The Second Institute of Oceanography, Ministry of Natural Resources of the People's Republic of China, Hangzhou 310012, China; wwding@sio.org.cn (W.D.); zhxk@sio.org.cn (X.Z.)
[*]   Correspondence: hangzhouwang@zju.edu.cn; Tel.: +86-0580-209-2203

**Abstract:** Floating ocean seismograph (FOS) is a vertical underwater vehicle used to detect ocean earthquakes by observing *P* waves at teleseismic distances in the oceans. With the requirements of rising to the surface and transmitting data to the satellite in real time and diving to the desired depth and recording signals, the depth control of FOS needs to be zero overshoot and accurate with fast response. So far, it remains challenging to implement such depth control due to the variation of buoyancy caused by the seawater density varying with the depth. The deeper the water is, the greater the impacts on buoyancy are. To tackle it, a fuzzy sliding mode controller considering the influence of seawater density change is proposed and simulated in MATLAB/SIMULINK based on the variable buoyancy system and state space function of FOS. Compared with proportional-integral-derivative (PID) controller, fuzzy PID controller and sliding mode controller, the simulation results indicate that the proposed controller shows its superiority regardless of the disturbing force. Its advantages include smaller steady-state error, faster response time, smaller system chatter, and well robustness. This proves that the designed fuzzy sliding mode controller is able to meet the working requirements and thus, lays a foundation for FOS application.

**Keywords:** floating ocean seismograph; variable buoyancy system; depth control; fuzzy proportional-integral-derivative (PID) control; fuzzy sliding mode control

---

## 1. Introduction

With the development of human science and technology, the exploration process of the ocean is gradually expanding to include deeper depths and more remote locations, and the research on the structure and the internal dynamics of the deep earth has become critical. The detection of ocean seismic waves serves to notice the thermal or compositional anomalies in Earth's mantle and core [1–3], and to image the mantle plume [4]. However, the lack of the ocean seismic stations poorly constrained the unveiling of the earth structure due to its 70% coverage by water. The formation of the ocean seismograph network is of great significance and value to scientific research [5], and floating ocean seismographs (FOS) can exactly realize such value [6]. It is a vertical underwater vehicle used to observe *P* waves (primary waves, the first signal from an earthquake) in ocean earthquakes [7,8].

Unlike the conventional stationary land seismic stations and submersible ocean bottom seismographs (OBS), FOS is suspended in the water column during its work time after diving to the desired parking depth set in the SOFAR (sound fixing and ranging channel) layer shown in

Figure 1 [1]. It is used to detect seismic waves and record seismic signals while drifting with ocean currents. It rises to the surface and transmits the data to the satellite when *P* waves are detected or the memory card is full. Then, it dives and another cycle starts. After a number of FOSs are densely deployed, they drift along with ocean currents with no need of a horizontal thruster or any other form of power engine and spread over the oceanic area, forming a global scope of seismic monitoring network [9]. Its global positioning system (GPS) makes the real-time transmission of data more convenient [2]. Its buoyancy control system controls its dive to the preset parking depth and float to the surface to communicate with satellite.

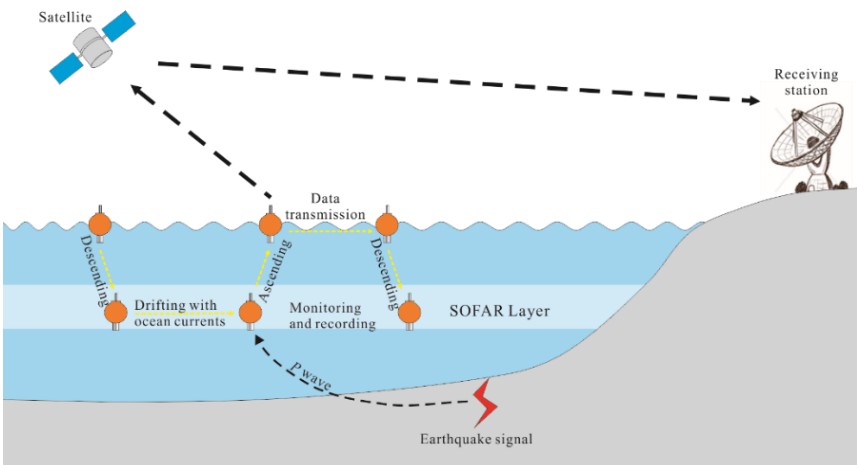

**Figure 1.** Working principle diagram of the floating ocean seismograph (FOS).

In the working process, FOS fluctuates with flows and currents at the water depth of 1000 m and may be subject to disturbance forces from flows, currents, sea animals, and other uncertain objects. This would affect the real depth of FOS resulting in lower measurement accuracy and even lead to bottoming out with unsatisfactory depth control. It is also known that as the depth of the FOS increases, its buoyancy varies due to the density change of seawater. While the variable buoyancy system (VBS) is used to adjust FOS drainage volume to change its buoyancy, mainly depending on its design is difficult to achieve the accurate depth control. Furthermore, to reduce the impact caused by the disturbing forces, the response performance of FOS under the disturbing forces should also be critical and be verified, as well. Therefore, to tackle the abovementioned challenges, this paper mainly proposes the design of a depth controller for FOS with fast response, zero overshoot, nearly no steady-state error, and high robustness, so that when FOS detects a seismic wave, it could respond accurately, rising to the surface and sending the data signal to satellite, and then quickly diving to the desired parking depth to resume the detection work without accidental bottoming damage. In addition, precise depth position can keep different FOSs at different depths so that they would not collide with each other in the future network control research.

At present, the existing basic control methods mainly include proportional–integral–derivative (PID) control [10], sliding mode control (SMC) [11], neural network control [12], predictive control [13], and fuzzy logic control (FLC) [14,15]. Their advantages and disadvantages are as shown in Table 1 [16].

With the consideration that FOS requires long-term steady work under water and the movement of diving and surfacing to transmit signals cannot be too slow, neural network control and predictive control are not considered because they need a long time to train and calculate. The PID control method is widely used in marine industrial products because of its simple structure and easy operation [17,18], the fuzzy logic control method is renowned for its simplicity, robustness, and anti-interference ability [14,19], and sliding mode control is useful for nonlinear system despite its tendency to cause "tremor" (small, rapid fluctuations in position). Two of these three control methods could be incorporated to control nonlinear systems such as FOS, and such controllers are mainly fuzzy PID

controller (F-PID) [20,21] and fuzzy sliding mode controller (F-SMC) [22,23]. To some extent, the fuzzy logic method can improve the weak points of PID control and sliding mode control, but we do not know which is better. To deal with the problem that which one of the controllers is the most suitable for the FOS depth control, different controllers should be designed and analyzed. In addition, the controller to be designed for FOS thereupon is required to be able to achieve the following four goals:

- Satisfy the depth control requirements, especially the ability to maintain position at a parking depth.
- Control FOS with sufficient stability, shorter adjustment time, smaller overshoot, and smaller steady-state error approaching zero while hovering in the sea.
- Have better dynamic characteristics compared with a traditional PID controller and a sliding mode controller (fuzzy logic control has certain steady-state error, so it is not taken into consideration).
- Have good anti-interference ability and robustness.

After a brief introduction, this paper is organized as follows. Section 2 analyzes the forces that FOS is subject to and derives its dynamic model as the controller design basis with the consideration of the influence of seawater density change on the buoyancy. Section 3 describes the design of different controllers for FOS depth control, including PID controller, fuzzy PID controller, sliding mode controller, and fuzzy sliding mode controller. In Section 4, the simulation comparisons and the analysis results are elaborated and conclusions are presented in Section 5.

**Table 1.** Advantages and disadvantages of different basic control methods.

| Types | | Advantages | Disadvantages |
|---|---|---|---|
| **Type** | **Acronym** | | |
| Proportional–integral–derivative | PID | Easy to implement and maintain | Mainly used for linear time invariant systems |
| Sliding mode control | SMC | Used for nonlinear systems | Cause actuator tremors, energy waste, and error proneness |
| Neural network control | NNC | Convergence to a precise model | Need longer training time and slower learning rate to get accurate model |
| Predictive control | PC | Good performance, strong robustness, low requirements for model accuracy | Heavy calculations, not suitable for fast time-varying systems |
| Fuzzy logic control | FLC | Simple control structure, efficient design, good robustness, strong anti-interference ability, no need for precise models | Lower accuracy and worse quality with simple fuzzy information, have steady-state errors |

## 2. Dynamic Models

Dynamic models are used to describe the regularity of changes in the system variables over time. In the modeling, the FOS needs to maintain its working depth of 1000 m in the water column, despite buoyancy changes due to seawater density variations, which has certain impacts on FOS when considering its diving and surfacing movement control. In this section, the structure of FOS seismograph was illustrated first and then its subjected forces were analyzed in which the hydrodynamic equation was established and the transfer function of depth to the vertical force were derived.

### 2.1. Coordinate Systems and Forces

FOS is mainly composed of glass shell, protective housing, iridium, acoustics module, conductivity-temperature-density (CTD), micro control panel, battery pack, VBS, and oil bladder. Its outside view is shown in Figure 2, in which the protective shell at two ends are open and the inner glass float is sealed with the vacuum sealing method. In addition, the devices, such as iridium and acoustics module, outside the glass float are relatively small compared to the glass float, and their volume could be therefore negligible in the forces analysis. In the calculation, FOS is assumed to be spherical, and its vertical plane motion coordinate system is shown in Figure 3. The axis is right-handed.

The general coordinate system (*E-ξηζ*) is fixed to the earth, and the local coordinate system (*G-xyz*) is fixed to the FOS. The coordinate variables of FOS in its local coordinate system are listed in Table 2.

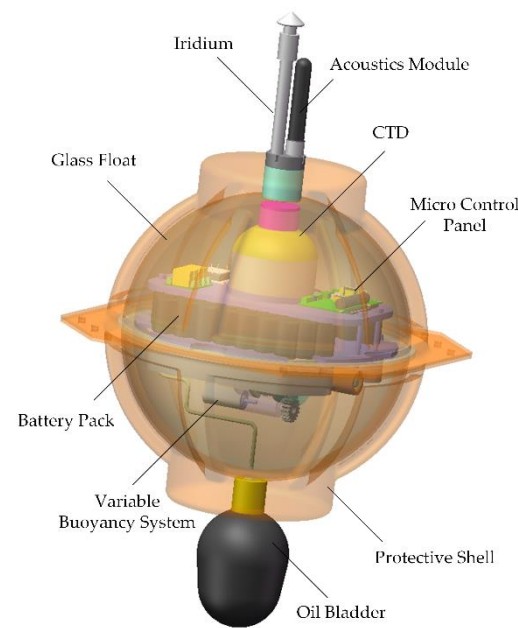

**Figure 2.** Outside view of floating ocean seismograph.

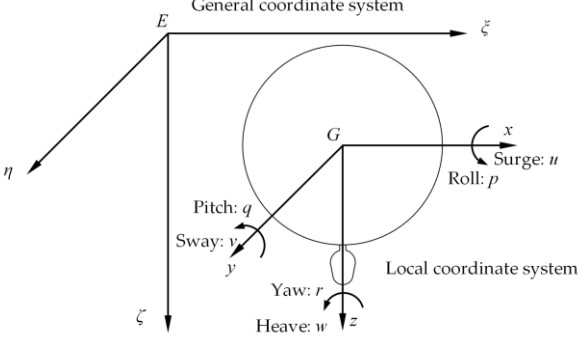

**Figure 3.** Vertical plane motion coordinate system.

**Table 2.** Variables of FOS in its local coordinate system.

| Degree of Freedom | Motion | Forces/Moments | Linear Velocity/Angular Velocity | Positions/Angles |
|---|---|---|---|---|
| 1 | Surge | $X$ (N) | $u$ (m/s) | $x$ (m) |
| 2 | Sway | $Y$ (N) | $v$ (m/s) | $y$ (m) |
| 3 | Heave | $Z$ (N) | $w$ (m/s) | $z$ (m) |
| 4 | Roll | $K$ (N·m) | $p$ (rad/s) | $\varphi$ (rad) |
| 5 | Pitch | $M$ (N·m) | $q$ (rad/s) | $\theta$ (rad) |
| 6 | Yaw | $N$ (N·m) | $r$ (rad/s) | $\psi$ (rad) |

FOS is mainly subject to two static forces, i.e., buoyancy and its own gravity. The gravity here is taken as the total displacement of FOS in water, denoted by $W$. Additionally, the total buoyancy $B$ here includes the full drainage volume buoyancy of FOS $B_0$, the changes of buoyancy caused by the variance of seawater density $B_\rho$, and the buoyancy provided by the variable buoyancy system $F_{VBS}$, as shown below:

$$B = B_0 + B_\rho + F_{VBS} \tag{1}$$

Here, the buoyancy variation $B_\rho$ is given by [24]:

$$B_\rho = (\rho(\zeta) - \rho_0)g\nabla \tag{2}$$

and the density $\rho(\zeta)$ is substituted by potential density as shown in Equation (3):

$$\rho(\zeta) = \sigma(\zeta) + 1000 \text{ kg}/\text{m}^3 \tag{3}$$

where $\rho(\zeta)$ is the density function varying with the depth (the data of conditional density $\sigma(\zeta)$ came from South China Sea observations in 2012) and could be expressed as Equation (4) by data fitting shown in Figure 4; $\zeta$ is the depth from sea surface in the general coordinate system; $\rho_0$ is the sea surface density; $\nabla$ is the full drainage volume of FOS.

$$\rho(\zeta) = \begin{cases} 1020.56 \text{ kg}/\text{m}^3 & 0 < \zeta < 66.37 \text{ m} \\ \left(26.13e^{4.674\times10^{-5}\zeta} - 11.36e^{-0.01052\zeta} + 1000\right) \text{ kg}/\text{m}^3 & \zeta \ge 66.37 \text{ m} \end{cases} \tag{4}$$

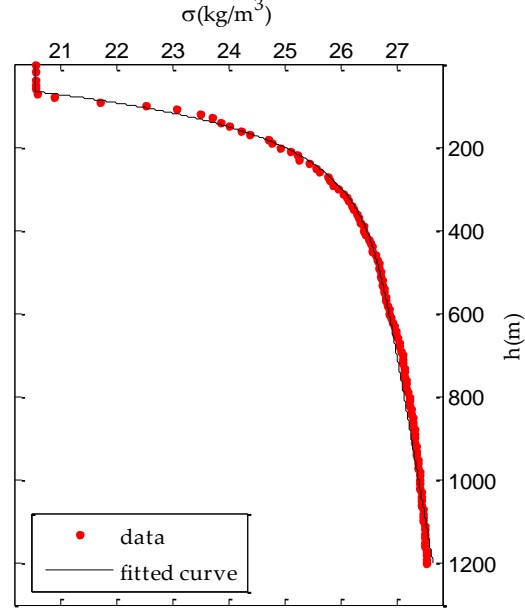

**Figure 4.** Seawater density fitting curves vary with depth at a point of South China Sea.

Accordingly, the variation of buoyancy caused by change of sea density $B_\rho$ can be described as:

$$B_\rho = \begin{cases} 0 & 0 < \zeta < 66.37 \text{ m} \\ \left(\rho(\zeta) - 1020.56 \text{ kg}/\text{m}^3\right)g\nabla & \zeta \ge 66.37 \text{ m} \end{cases} \tag{5}$$

In addition, the buoyancy provided by the variable buoyancy system $F_{VBS}$ can be expressed as:

$$F_{VBS} = \int_0^{t_1} \rho(\zeta)gq_1 dt - \int_0^{t_2} \rho(\zeta)gq_2 dt \tag{6}$$

where $q_1$ is the volume flow rate while pumping oil from the bladder to the tank and $q_2$ is the volume flow rate while pumping oil in the opposite direction, and therefore these two parts cannot coexist. The values of $t_1$ and $t_2$ depend on the maximum buoyancy adjustment volume and volume flow rate of VBS.

When balancing counterweights, the gravity $W$ is equal to the full drainage volume buoyancy $B_0$. Therefore, the residual static load of FOS $\triangle P$ can be obtained:

$$\Delta P = W - B = W - \left( B_0 + B_\rho - F_{VBS} \right) = F_{VBS} - B_\rho \tag{7}$$

When $\triangle P > 0$, FOS sinks; when $\triangle P < 0$, FOS rises up; when $\triangle P = 0$, FOS is in buoyancy equilibrium.

*2.2. Dynamic Models*

For depth control of FOS, its six degree of freedom hydrodynamic equation should be established. The well-known hydrodynamic equation [25] can be simplified as equation (8) for FOS [26] with assumptions and specific simplifications as shown below:

- Only the motion of FOS in the vertical plane is considered due to the particularity of movement of the ocean current. Then, the movement state variables in Table 2 can be simplified as $u$, $w$, and $q$, namely the horizontal speed, the vertical speed, and the angular speed around the y-axis, respectively. In addition, the value of sway, pitch, and yaw is zero $v = p = r = 0$.
- Higher-order damping, second-order terms in $u$ and $q$ are neglected in the motion of FOS. The horizontal speed, the angular speed around the y-axis, the horizontal acceleration, and the angular acceleration around the y-axis are assumed as zero for convenience, that is $u = q = \dot{u} = \dot{q} = 0$.
- Buoyancy equal to weight ($B = W$) at the initial time. Two centers of gravity and buoyancy coincide with each other. Only the disturbance force in z-direction $d$ is considered.

$$\left( m - Z_{\dot{w}} \right) \dot{w} - Z_w w = Z + d \tag{8}$$

where $m$ is mass of FOS; $w$ is vertical speed; $\dot{w}$ is vertical accelerate; $Z_{\dot{w}}$ denoting partial derivative of vertical force with respect to vertical acceleration is approximately the additional mass of a ball, and its formula is shown in Equation (9); $Z_w$ denoting partial derivative of vertical force with respect to vertical speed is calculated as Equation (10) according to the ITTC nondimensional physical quantity definition [27]; $Z$ is the vertical force, and $Z = \triangle P$; $d$ is a sinusoidal disturbance force.

$$Z_{\dot{w}} = -\frac{1}{2}\rho \cdot \frac{4}{3}\pi R^3 = -\frac{2}{3}\rho\pi R^3 \tag{9}$$

$$Z_w = \frac{1}{2}\rho(2R)^2 U Z'_w = 2\rho R^2 U Z'_w \tag{10}$$

In the assumption, the value of roll, pitch, and yaw is zero, so $z = \zeta$ and the variable $\zeta$ in Equations (2)–(6) can be replaced by $z$. Additionally, because the derivative of depth to time is equal to the vertical velocity $\dot{z} = w$, Equation (8) can be transformed as:

$$\ddot{z} = \frac{Z_w}{m - Z_{\dot{w}}}\dot{z} + \frac{1}{m - Z_{\dot{w}}}\left( F_{VBS} - B_\rho + d \right) \tag{11}$$

Therefore, the state-space equation is given by:

$$\begin{bmatrix} \dot{z} \\ \ddot{z} \end{bmatrix} = \begin{bmatrix} 0 & 1 \\ 0 & \frac{Z_w}{m-Z_{\dot{w}}} \end{bmatrix} \begin{bmatrix} z \\ \dot{z} \end{bmatrix} + \begin{bmatrix} 0 \\ \frac{1}{m-Z_{\dot{w}}} \end{bmatrix} \left( F_{VBS} - B_\rho + d \right) \tag{12}$$

In Equations (9)–(10), the density $\rho$ is approximately constant for convenience of calculation. According to the hydrodynamic coefficients of round disc underwater vehicles and ellipsoidal underwater vehicles [28,29], the hydrodynamic coefficient of this spherical FOS is estimated as $Z'_w = -0.4$. In addition, the numerical values of the major parameters in the calculation are listed in Table 3.

**Table 3.** The major parameters in the calculation.

| Parameters | Numeric Values | Physical Significances |
|---|---|---|
| $\rho$ | 1025 kg/m$^3$ | Density of sea water |
| $m$ | 43.20 kg | Mass of FOS |
| $R$ | 0.216 m | Radius of floating ocean seismograph |
| $U$ | 0.8 m/s | Freestream velocity |
| $V$ | 1.5 L | Maximum buoyancy adjustment volume |
| $q_1, q_2$ | $1.5 \times 10^{-4}$ m$^3$/s | Volume flowrate of VBS |

## 3. Depth Controller Design

### 3.1. Depth Control Principle

The depth control of FOS is realized by a variable buoyancy system and a depth controller, and the control block diagram thereof is shown in Figure 5. In the diagram, the input of the system is the desired depth $z_d$, and the inputs of the depth controller are the depth error $e$ and the depth error change rate $\dot{e}$ of FOS. The output of the depth controller is the expected buoyancy $B_{exp}$ after the adjustment, and the system output is the actual depth $z$ of FOS derived from the variable buoyancy system and FOS dynamic system function. The variable buoyancy system is an actuator for FOS. The depth sensor serves as a feedback for measuring the depth, and the controller continuously adjusts the variable buoyancy system according to the feedback information to reach the target depth. At the meantime, the seawater density at actual depth is fed back to the controller in real time, which allows the more accurate buoyancy adjustment.

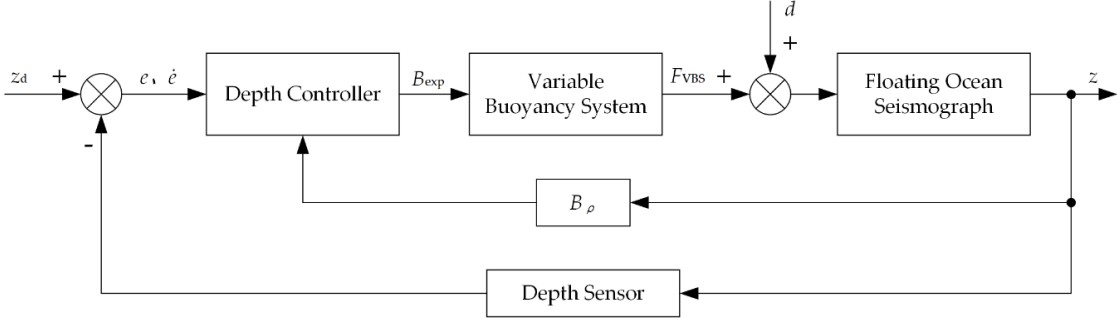

**Figure 5.** Depth control block diagram.

The expression equation of the depth error $e$ is as follows:

$$e = z_d - z \tag{13}$$

Different depth controllers are proposed in Sections 3.2–3.5, including PID controller, fuzzy PID controller, sliding mode controller, and fuzzy sliding mode controller. The PID controller and sliding mode controller are enhanced into fuzzy PID controller and fuzzy sliding mode controller, respectively. Their design and analysis are as follows.

### 3.2. PID Controller

The block diagram of PID controller is shown in Figure 6. The control input of this controller is merely the depth deviation $e$. The control output of the PID controller is the expected buoyancy $B_{exp}$ given by Equation (14):

$$B_{exp} = K_{P1} + K_{I1} \int e \mathrm{d}t + K_{D1} \frac{\mathrm{d}e}{\mathrm{d}t} + B_\rho \tag{14}$$

where $K_{P1}$, $K_{I1}$, and $K_{D1}$ are proportional, integral, and differential coefficients, respectively.

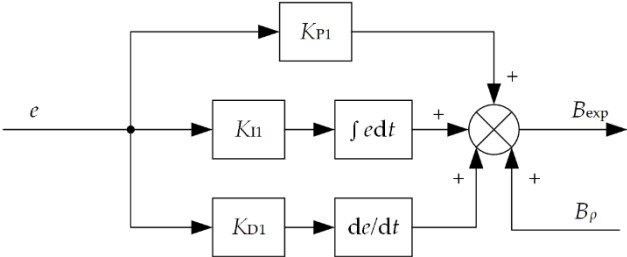

**Figure 6.** Proportional–integral–derivative (PID) depth controller block diagram.

### 3.3. Fuzzy PID Controller (F-PID)

To enhance the PID controller, the fuzzy logic method is used to allow time-variable tuning of proportional, integral, and differential coefficients, which is fuzzy PID controller. The block diagram of the fuzzy PID controller is as shown in Figure 7 [16,30], in which the controller parameters $K_P$, $K_I$, and $K_D$ can be calculated as:

$$\begin{cases} K_P = K_{P0} + \alpha \Delta K_P \\ K_I = K_{I0} + \beta \Delta K_I \\ K_D = K_{D0} + \gamma \Delta K_D \end{cases} \tag{15}$$

where $K_{P0}$, $K_{I0}$, and $K_{D0}$ are the initial values of PID parameters; $\alpha$, $\beta$, and $\gamma$ are correction factors which can be adjusted according to the actual control effect to lower the requirements for accuracy from the fuzzy control rule tables [30–32]; $\triangle K_P$, $\triangle K_I$, and $\triangle K_D$ are adjusting parameters used for regulating the values of $K_{P0}$, $K_{I0}$, and $K_{D0}$ within a certain range.

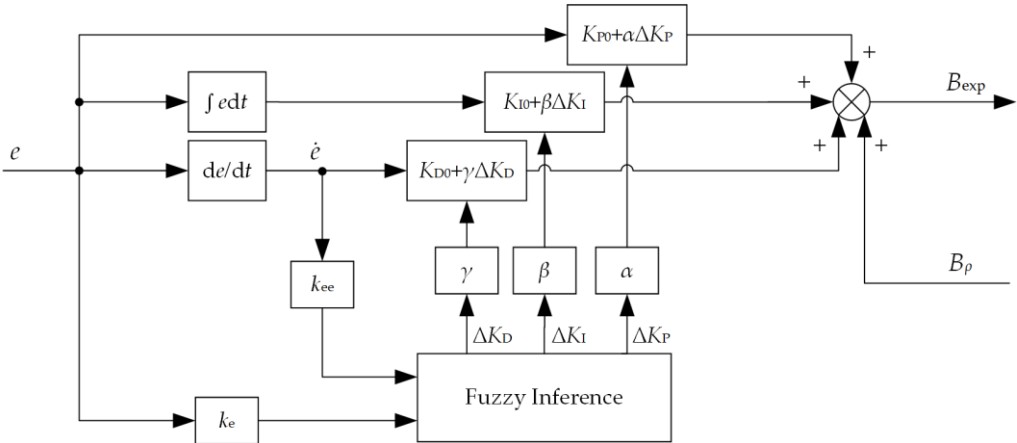

**Figure 7.** Fuzzy PID depth controller block diagram ($k_e$ and $k_{ee}$ are adjustment for fuzzy inference inputs; $\alpha$, $\beta$, and $\gamma$ are correction factors for fuzzy inference outputs).

The self-adaptive fuzzy PID controller takes error $e$ and the depth error change rate $\dot{e}$ as inputs. $k_e$ and $k_{ee}$ are adjustment factors used for regulating the range of $e$ and $\dot{e}$, respectively to meet the input requirements of fuzzy inference module. The controller modifies the PID parameters online using the fuzzy inference to realize the self-adaptive adjustment requirements for the PID parameters $K_P$, $K_I$, and $K_D$ with different values of $e$ and $\dot{e}$. Therefore, its controlled object will have a good dynamic and static performance with anti-interference ability. The fuzzy inference is constructed by the fuzzy rules and the membership functions. The output of the controller can be described as:

$$B_{\exp} = K_P + K_I \int edt + K_D \frac{de}{dt} + B_\rho = K_{P0} + \alpha \Delta K_P + (K_{I0} + \beta \Delta K_I) \int edt + (K_{D0} + \gamma \Delta K_D) \frac{de}{dt} + B_\rho \tag{16}$$

The inputs of the fuzzy inference module are the depth error $e$ and the depth error change rate $ec$, and their fuzzy domain is $[-4, 4]$. The outputs are $\triangle K_P$, $\triangle K_I$, and $\triangle K_D$, and their fuzzy domain is $[-3, 3]$.

Then, the fuzzy domains above use the same fuzzy subset which are negative big (NB), negative middle (NM), negative small (NS), zero (ZO), positive small (PS), positive middle (PM), and positive big (PB). Additionally, the types of membership function of input and output variables are Gaussian and triangular, respectively [26]. Mamdani-type fuzzy rules are adopted, of which the specifics are shown in Table 4 [20,33,34]. An example is provided here with highlight. When $e \cdot k_e$ is characterized as the negative middle and $\dot{e} \cdot k_{ee}$ is characterized as the positive middle, then the corresponding table entry ZO/ZO/ZO indicates that zero values are returned for $\triangle K_P$, $\triangle K_I$, and $\triangle K_D$, respectively.

**Table 4.** Fuzzy control rules for $\triangle K_P$, $\triangle K_I$, and $\triangle K_D$ with inputs $e \cdot k_e$ and $\dot{e} \cdot k_{ee}$.

| $\triangle K_P/\triangle K_I/\triangle K_D$ | $\dot{e} \cdot k_{ee}$ | | | | | | |
|---|---|---|---|---|---|---|---|
| $e \cdot k_e$ | **NB** | **NM** | **NS** | **ZO** | **PS** | **PM** | **PB** |
| NB | PB/NB/PB | PB/NB/PB | PM/NM/PM | PM/NM/PM | PS/NS/PS | PS/ZO/ZO | ZO/ZO/ZO |
| NM | PB/NB/PB | PM/NB/PB | PM/NM/PM | PS/NS/PS | PS/NS/PS | ZO/ZO/ZO | NS/ZO/ZO |
| NS | PM/NB/PB | PM/NM/PM | PS/NS/PS | PS/NS/PS | ZO/ZO/ZO | NS/PS/NS | NS/PS/NS |
| ZO | PM/NM/PM | PS/NM/PM | PS/NS/PS | ZO/ZO/ZO | NS/PS/NS | NS/PM/NM | NM/PM/NM |
| PS | PS/NM/PM | PS/NS/PS | ZO/ZO/ZO | NS/PS/NS | NS/PS/NS | NM/PM/NM | NM/PB/NB |
| PM | PS/ZO/ZO | ZO/ZO/ZO | NS/PS/NS | NS/PS/NS | NM/PM/NM | NM/PB/NB | NB/PB/NB |
| PB | ZO/ZO/ZO | NS/ZO/ZO | NS/PS/NS | NM/PM/NM | NM/PM/NM | NM/PB/NB | NB/PB/NB |

## 3.4. Sliding Mode Controller (SMC)

In the process of designing the sliding mode control law, the sliding surface $s$ is defined as:

$$s = \lambda e + \dot{e} \tag{17}$$

where $\lambda$ is a positive constant.

The output of the controller is designed based on Equation (11) as:

$$B_{\exp} = \left( m - Z_{\dot{w}} \right) \left[ \lambda \dot{e} + \ddot{z}_d - \frac{Z_w}{m - Z_{\dot{w}}} \dot{z} + \eta \mathrm{sgn}(s) \right] + D\mathrm{sgn}(s) + B_\rho \tag{18}$$

where $\eta$ is a positive constant; $D$ is the upper bound of $|d|$, that is $|d| \leq D$.

For the stability analysis, the Lyapunov function $V$ is chosen as:

$$V = \frac{1}{2}s^2 \tag{19}$$

Therefore, combined with Equations (12)~(13), we can get:

$$\dot{V} = s\dot{s} = s\left( \lambda \dot{e} + \ddot{e} \right) = s\left( \lambda \dot{e} + \ddot{z}_d - \ddot{z} \right) = s\left[ \lambda \dot{e} + \ddot{z}_d - \frac{Z_w}{m - Z_{\dot{w}}} \dot{z} - \frac{1}{m - Z_{\dot{w}}} \left( F_{VBS} - B_\rho + d \right) \right] \tag{20}$$

Replace $F_{VBS}$ with controller output $B_{\exp}$, Equation (20) can be written as:

$$\dot{V} = s\left( -\eta\mathrm{sgn}(s) - \frac{1}{m - Z_{\dot{w}}}d - \frac{1}{m - Z_{\dot{w}}}D\mathrm{sgn}(s) \right) = -\eta|s| - \frac{1}{m - Z_{\dot{w}}}\left( ds + D|s| \right) \leq -\eta|s| \leq 0 \tag{21}$$

According to the Lasalle invariance principle, when $\dot{V} \equiv 0, s \equiv 0$, and the system is asymptotically stable. That is to say, when $t \to \infty$, $s \to 0$, and its convergence speed depends on $\eta$. Additionally, for the controller, the larger the value of $D$, the better the robustness, but it will also increase the tremor of the system.

## 3.5. Fuzzy Sliding Mode Controller (F-SMC)

The parameter $D$ in sliding mode controller may lead to trembling, so the fuzzy logic method is used to adjust the value of $D$ to reduce the tremor caused by sliding mode controller. The schematic

of implementation of fuzzy sliding mode controller is shown in Figure 8, of which the sliding mode control law is established in the same way as Section 3.4.

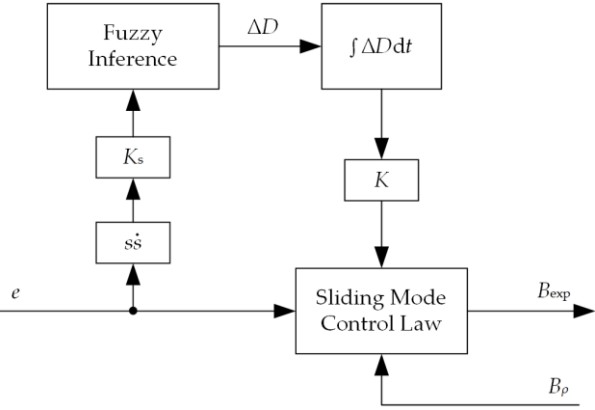

**Figure 8.** Block diagram of fuzzy sliding mode depth controller.

The output of the controller is then given as:

$$B_{\exp} = \left(m - Z_{\dot{w}}\right)\left[\lambda\dot{e} + \ddot{z}_d - \frac{Z_w}{m - Z_{\dot{w}}}\dot{z} + \eta\,\mathrm{sgn}(s)\right] + \hat{D}\,\mathrm{sgn}(s) + B_\rho \tag{22}$$

$$\hat{D} = K\int_0^t \Delta D\,\mathrm{d}t \tag{23}$$

where *K* is a gain factor; $\triangle D$ depends on fuzzy inference.

The stability analysis of the controller is the same as the sliding mode controller, so it is omitted.

When the system reaches the sliding surface, it will stay on it only if $s\dot{s} < 0$. In other words, for the fuzzy rules: If $s\dot{s} > 0$, $\Delta D$ should increase; if $s\dot{s} < 0$, $\Delta D$ should decrease. The fuzzy inference is then designed based on these rules, and its input can be adjusted by the scale factor $K_s$.

The input of the fuzzy inference module is $s\dot{s}$, and its fuzzy domain is [-4, 4]. The output of the fuzzy inference module is $\triangle D$, and its fuzzy domain is [-8, 8]. The fuzzy subsets used for fuzzy domains both are (NB, NM, NS, ZO, PS, PM, PB). The types of membership function of fuzzy input and output are shown in Figure 9, and the fuzzy rules are shown in Table 5 using Mamdani-type.

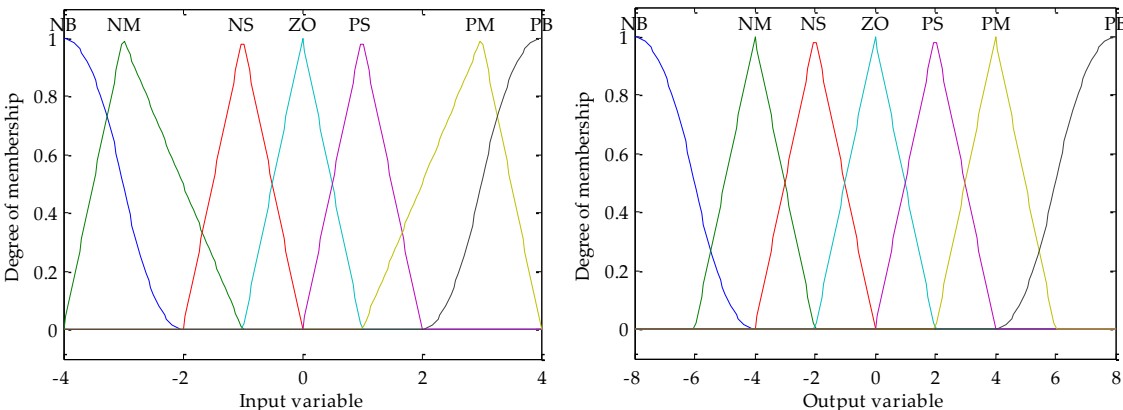

**Figure 9.** Types of membership function of input and output variables.

**Table 5.** Fuzzy control rules for input variable $s\dot{s}$.

| $s\dot{s}$ | NB | NM | NS | ZO | PS | PM | PB |
|---|---|---|---|---|---|---|---|
| $\triangle D$ | NB | NM | NS | ZO | PS | PM | PB |

## 4. Simulation Results and Analysis

The different depth controllers designed in Section 3 are implemented and simulated in MATLAB/SIMULINK, and their performance is compared with each other.

To analyze the effect of seawater density change, the PID controller is used to compare the two cases with and without the variation of buoyancy caused by the change of seawater density $B_\rho$, and the response curves are shown in Figure 10.

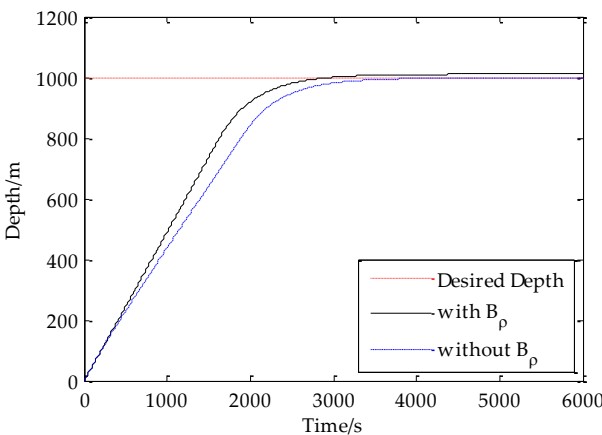

**Figure 10.** Depth response of PID depth controller.

This figure shows that when the depth is lower than 200 m, whether considering the influence of seawater density has little effect on the response result; while the depth exceeds 200 m, the response time with considering the influence of seawater density is significantly faster than that without consideration of $B_\rho$. Therefore, subsequent simulations are implemented considering the variation of buoyancy caused by the change of sea density $B_\rho$.

### 4.1. Simulation and Analysis without Disturbing Force

The response of the system can be observed by inputting a step signal with an amplitude of 1000 signifying the target parking depth. The processes of FOS first diving from the surface (z = 0 m) to 1000 m and then suspending are simulated. For discussion, the results are compared with each other based on four performance parameters, which is the rise time, settling time, steady-state error, and maximum overshoot. The descriptions of different controllers are listed in Table 6.

**Table 6.** Parameters of different controllers.

| Controller Types | Control Parameters |
|---|---|
| PID | $K_P = 0.2$, $K_I = 0.000002$, $K_D = 60$ |
| F-PID | $K_e = 0.018$, $K_{ee} = 0.4$, $K_{P0} = 0.2$, $K_{I0} = 0.000002$, $K_{D0} = 60$, $\alpha = 0.2$, $\beta = 0.00002$, $\gamma = 20$ |
| SMC | $\lambda = 0.02$, $D = 15$, $\eta = 0.5$ |
| F-SMC | $\lambda = 0.02$, $\eta = 0.5$, $K_s = 0.000001$, $K = 0.8$ |

The depth control response curves of FOS with PID, fuzzy PID, sliding mode, and fuzzy sliding mode controllers are shown in Figure 11. It indicates that the depth response curves of the four control

methods have almost no oscillation and tend to be stable. Approximately from 0–1500 s, the curves of the four controllers almost coincide due to the maximum volume flow rate of VBS. The velocity curves based on the four controllers are shown in Figure 12. It can be seen from the figure that the sinking speeds of the FOS with different controllers are 0.50 m/s (PID, Fuzzy PID), 0.49 m/s (SMC, Fuzzy SMC), respectively. For further analysis, characteristic response parameters obtained from the simulation results are presented in Table 7. Since the depth curves of SMC and fuzzy SMC methods do not reach 1000 with the steady-state error of -0.2 m, their maximum overshoots are approximately equal to 0.

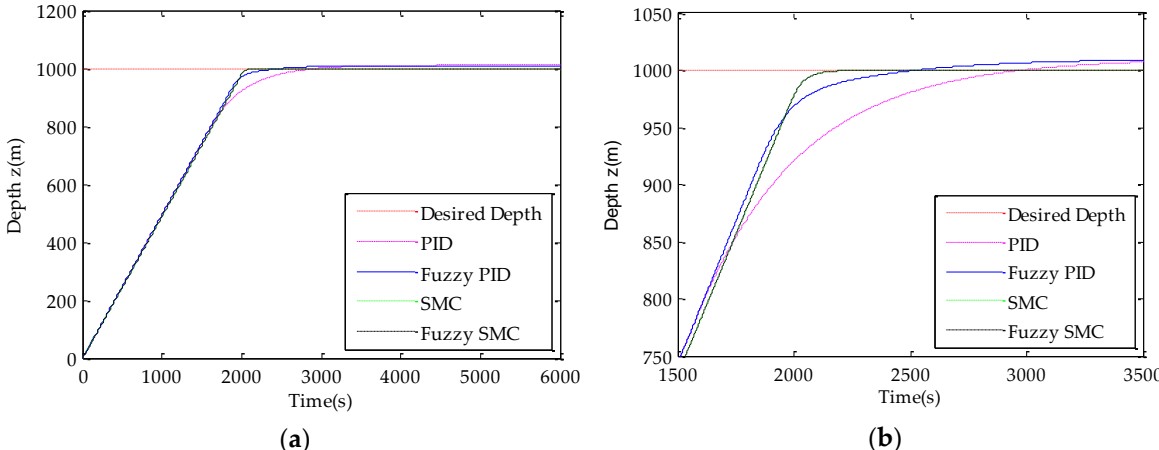

**Figure 11.** Control response curves without disturbing force. (**a**) Depth response curves. (**b**) Magnified view of depth response curves.

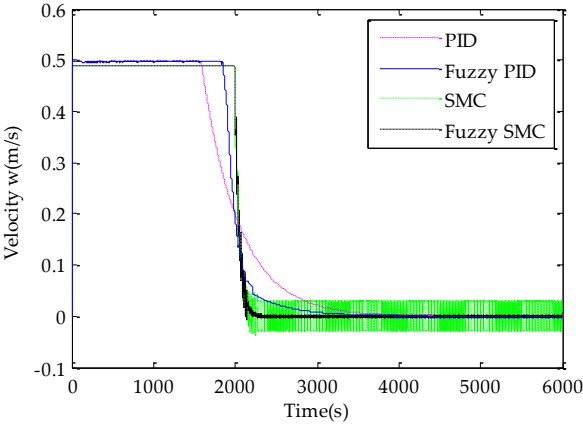

**Figure 12.** Velocity curves of depth control without disturbing force.

**Table 7.** Characteristic parameter table of four controllers.

| Controller Types | Rise Time (s) | Settling Time (s) | Steady-State Error (m) | Maximum Overshoot (%) |
|---|---|---|---|---|
| PID | 1702.0 | 2671.5 | 10.2 | 1.027 |
| F-PID | 1610.6 | 2213.2 | 9.9 | 0.996 |
| SMC | 1632.7 | 2040.1 | -0.2 | $\approx 0$ |
| F-SMC | 1632.7 | 2040.0 | -0.1 | $\approx 0$ |

Compared with the traditional PID controller, the fuzzy PID controller has better performance with the rise time reduced by 5.37%, the settling time decreased by 17.16%, the steady-state error decreased by 2.94%, and the maximum overshoot decreased by 3.02%. Compared with the sliding mode controller, the characteristic parameters of fuzzy sliding mode controller change a little. However,

its velocity is more stable than the SMC method, which greatly saves its energy consumption. These indicate that in the depth control of FOS, the performance of fuzzy PID controller and fuzzy sliding mode controller is significantly better than pure PID controller and SMC controller. Furthermore, except for the rise time, the fuzzy sliding mode controller has shorter settling time (decreased by 7.8%), smaller steady-state error (decreased by 99.02%), and lower maximum overshoot compared with the fuzzy PID controller. The conclusion can be drawn that the designed fuzzy sliding mode controller has the advantages of fast response, high precision, and excellent steady-state performance without disturbing force.

### 4.2. Simulation and Analysis with Disturbing Force

To analyze the anti-interference ability of the designed controller, a disturbing force as Equation (24) is added to the system. The response curves of the four controllers in the presence of disturbing force are shown in Figure 13, of which the corresponding characteristic parameters are shown in Table 8 (due to the existing of the disturbing force, the response curves are fluctuating and the steady-state error does not exist.).

$$d = 15\sin(\pi t/2) \tag{24}$$

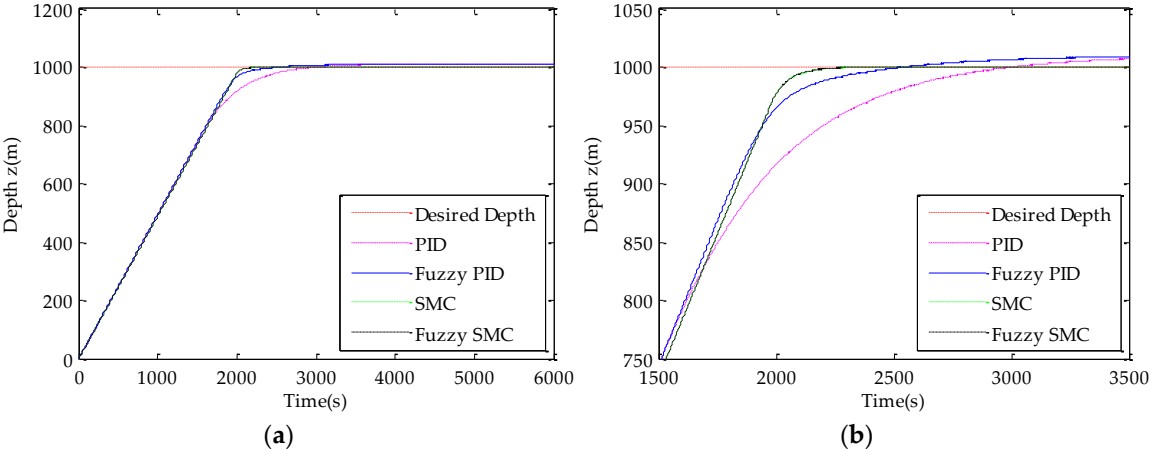

**Figure 13.** Control response curves with disturbing force. (**a**) Depth response curves. (**b**) Magnified view of depth response curves.

**Table 8.** Characteristic parameter table after adding interference to the four controllers.

| Controller Types | Rise Time (s) | Settling Time (s) | Maximum Overshoot (%) |
|:---:|:---:|:---:|:---:|
| PID | 1726.1 | 2693.9 | 1.057 |
| F-PID | 1615.3 | 2233.9 | 1.007 |
| SMC | 1633.3 | 2061.3 | 0.013 |
| F-SMC | 1632.6 | 2061.4 | 0.013 |

In general, the parameters of these four control methods have changed while adding the disturbing force. To further compare the performance of the four control methods in the presence of disturbing force, the characteristic parameters without external disturbing force are subtracted from that with disturbing force, and the change of each parameter is shown in Table 9.

**Table 9.** Changes of characteristic parameters.

| Controller Types | Rise Time (s) | Settling Time (s) | Maximum Overshoot (%) |
|---|---|---|---|
| PID | +24.1 | +22.4 | +0.030 |
| F-PID | +4.7 | +20.7 | +0.011 |
| SMC | +0.6 | +21.2 | +0.013 |
| F-SMC | -0.1 | +21.4 | +0.013 |

It shows that under the same conditions of disturbing force, the changes of parameters for fuzzy PID control, SMC, and fuzzy SMC are less than that for the PID control. While the changes of settling time have only slight differences among the four controllers, the change of rise time for the fuzzy SMC method is the littlest. Additionally, its maximum overshoot is always small. Therefore, the designed fuzzy SMC controller shows better anti-interference capability and robustness and has better dynamic performance.

## 5. Conclusions

For the depth control of FOS, its mathematical model has been obtained based on the hydrodynamic equation and on the characteristic parameters of the buoyancy control device with considering the variation of seawater density with depth. By simulating the processes of diving and suspending in the cases with and without disturbing force, the designed fuzzy sliding mode controller has been compared with a traditional PID controller, a fuzzy PID controller, and a sliding mode controller. The results demonstrate that in both cases, the fuzzy sliding mode controller could achieve the expected depth in the nearly shortest time with the minimum steady-state error and overshoot, which indicates its rapid response capability, good stability, and high accuracy. In addition, its change of rise time caused by disturbing force is smaller than the other three controllers. This indicates its possession of good robustness and anti-interference ability. Moreover, the consideration of the changes of seawater density with depth in the design of the depth controller for FOS to enable the adaptive adjustment in real time is a highlight for the controller design. Therefore, we carefully draw the conclusion that the proposed fuzzy sliding mode controller could satisfy the depth control requirements of FOS.

In the application of depth control for the similar underwater vehicle, if a PID controller must be used, the fuzzy logic method can be implemented with it as fuzzy PID to reach the stable depth faster. If a sliding mode controller is able to be used, it is better than PID and F-PID, but it cannot guarantee the stability of the actuator. If good depth response performances and actuator stability are required at the same time, the fuzzy sliding mode controller is much better. However, in our research, there remains some issues here that the anti-interference performance of the FOS is still not optimal, so an observer will be considered in our next step work, and the experiments in South Sea shall be carried out to test FOS performances.

**Author Contributions:** Conceptualization, H.H. and H.W.; methodology, C.Z.; software, C.Z.; validation, W.D., Z.X., and G.S.; formal analysis, C.Z.; investigation, H.H. and C.Z.; resources, W.D. and X.Z.; data curation, G.S.; writing—original draft preparation, C.Z. and H.W.; writing—review and editing, H.H. and H.W.; supervision, H.H.; project administration, X.Z.; funding acquisition, H.H. and W.D. All authors have read and agreed to the published version of the manuscript.

**Funding:** This work was supported financially by the State Oceanic Administration Program on Global Change and Air-Sea Interaction (No. GASI-02-SHB-15).

**Acknowledgments:** This work was supported by the HPC Center of Zhejiang University (Zhoushan Campus).

**Conflicts of Interest:** The authors declare no conflict of interest.

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
