# Peer review of "Design of the Depth Controller for a Floating Ocean Seismograph"

_jmse, doi:10.3390/jmse8030166_

Round 1
Reviewer 1 Report
Summary of Paper.
Depth control is an important issue in the design of an autonomous underwater instrument, and is subject to unique engineering issues associated with the need to conserve power and to prevent damage to the vehicle.
This paper compares several types of depth controllers, the PID controller and the SLC controller, and their fuzzy logic enhancements, the F-PID anf F-SLC. Overall, I would summarize the Results as:
(1) PID and F-PID both lead to static overshoot or about the same amount, but F-PID reaches the stable depth faster.
(2) SMC and F-SMC have almost identical performance and are better than either PID and F-PID.
So the conclusions of the paper can be summarized as:
If you must use a PID controller, implement one with fuzzy logic, because although both overshoots, the F-PID reaches the stable depth faster. If, on the other hand, you can use a SMC controller, then do so, for it performs better than PID controller. Fuzzy logic does not significantly improve the performance of a SMC controller.
-----------------------------
Reviewer’s analysis:
(1) The results, while somewhat disappointing in that they demonstrate only modest advantage of fuzzy logic, are worth publishing, as they can inform the choice of depth-control systems used in future autonomous underwater instruments.
(2) The few simulations that are performed are marginally sufficient to analyze the performance of the controllers. However, the paper would be improved by more and various tests. For instance, little is said about the choice of the fuzzy logic categories NB … ZO … PB. Might a different choice have led to better performance. One could imagine addressing this issue through Monte Carlo methods: try many different category boundaries to find the optimum one.
(3) The conclusions are stated in a modest and forward-looking way and mostly correspond to my summary, above, but could be made clearer.
(4) The paper is full of undefined variables (ans especially k-sun-e and k-sub-ee), poorly-described operations, poorly (and incorrectly) labeled table captions and header, and uninformative figure captions and would be greatly improved by some very careful editing. See my stream-of-consciousness comments at the end of this review for specific suggestions.
---------------
Reviewer Recommendation:
As it stands, the paper appears “correct”, makes a “modest” contribution but needs substantial editing to improve presentation. The paper may be published after moderate revisions. The results might be made stronger by a greater number and variety of simulations, perhaps conducted via Monte Carlo methods.
---
Stream-of-Consciouness remarks:
Line 33, “gradually deepening” -> “gradually expanding to include deeper depths and more remote locations”
Line 42, “FOS is suspending during” -> “FOS is suspended in the water column during”
Line 69, “PID” -> “proportional–integral–derivative (PID)”
Line 77, Not sure what “despite his shortcoming of tremor” means. Maybe; “despite its tendency to cause “tremor” (small, rapid fluctuations in position)”.
Line 85, “especially its status of suspending” -> “especially the ability to maintain position”
Line 91, “while adding the disturbing force”. Premature to talk about forces her. Pend to later?
Line 92, Table 1. Break out “types” column into two, “type” and “acronym”
Line 101, “the working depth of FOS is 1000 m and the buoyancy changes” -> “The FOS needs to maintain its working depth of 1000 m in the water column, despite buoyancy changes due to seawater density variations.
Line 156, Equation 8. The use of tau to represent control forces is a bit confusing, because tau is so often used for stress.
Line 156, Equation 8. The introduction of the stochastic term bold-d is a bit misleading, as the analysis does not seem to be headed in a stochastic direction.
(Added later) I see that in the simulation, examples of a deterministic sinusoidal d are considered. This would be worth mentioning here.
Line 165, Equation 9, Symbols like Z-sub-w and Z-sub-w-dot are very hard to distinguish. Please use different notation.
Line 166, The X, Y, Z and M symbols in (9) need to be defined.
Section 2.2. I would be inclined to cut all the discussion of the general equation (8) and just start the paragraph asserting that (14) is simplification of the well-known hydrodynamic equation (reference) and then give in words the list of simplifications, i.e. rotations and horizontal motions are not being considered. At this point, ALL the variables in (14) need to be defined.
Line 172, Equation 11, Having variables R (radius) and L (diameter = 2R) only adds to confusion. Stick to radius. Break our volume from equation 11A, - (1/2) (rho) x (4/3)piR^3.
Line 172, Equation 11, Equation 11B must have a typo, for Z-sub-w appears on both left and right.
Line 171, I am not sure why tau-sub-z is being said at all. It is to show that stress is being defined as having the same sign as pressure, as contrasted to the opposite sign as is often done in continuum mechanics?
Line 180, “the density changes with depth are ignored for convenience of calculation”. Not sure what thus means. It would sound to me that it’s saying that a term involving the time derivative of density is being ignored. However, looking at (13), I see no way for such a term to arise. Clarify?
Line 184, Table 3, I can see that density is specified to be constant, but I am not sure why, since density fluctuations were referred to earlier. This issue is clearly connected to my previous comment.
Line 188, “mainly realized” -> “realized”
Line 208, Equation 16, B-sub-rho needs to be defined.
Line 213. Paragraph needs an opening sentence that indicates that the design of the PID controller is being enhanced into a Fuzzy PID controller by allowing time-variable tuning of the K-coefficients. At this point, I am wondering whether the FLC mentioned in the introduction is the F-PID, or whether it is just an intermediate step towards its description.
Line 221, Figure 7, At this point, the parameters k-sub-e and k-sub-ee (the input parameters of the fuzzy inference module) have not yet been defined in the text, so they must be defined here in the Table caption. These variables are of critical importance and need to be emphasized.
Line 222,” k-sub-e and k-sum-ee are adjustment factors of e and e respectively”. This in not an adequate definition of these two variables and needs to be substantially expanded.
Line 237, Table 4. This table needs to be much better described. I think that k-sub-e varies between rows and k-sub-ee bewteen columns (or maybe vice versa) but this is not apparent from the labels. The categorization “NB, NM …” I guess means “negative-big, negative-medium, …” I guess double appearance of NB in column and row headers is a typo and should be NS. The table caption needs to be significantly expanded.
Line 237, My inclination would be to highlight row 2 and column 5 and then provide an example, when z-sub-e is characterized as positive- small and z-sub-ee is characterized as negative-medium, then corresponding table entry ZO/ZO/ZO indicates that zero values are returned for â–³KP, â–³KI, and â–³KD, respectively.
Sections 3.4 and 3.5. Ah, now I see that the SLC is being enhanced into a F-SLC, just like the PID was inhanced into a F-PID. I am, of course, wondering whether yet more are to come. Looking back to Line 202, I can see that some attempt has been made to say this. I recommend that that sentence be expanded into a short paragraph.
Line 243, Equation 20. Some physical interpretation of this equation is neeed.
Line 275, Figure 9. Is this categorization rule specific to the F-SLC, or does it apply to the F-PID as well? If the latter, it needs to be moved up.
Line 278. I don’t recall B-sub-rho ever being defines, but in any case its definition and physical interpretation should be re-stated here.
Line 287. The with B-sub-rho case overshoots. Is this really correct? Or are the curves mis-identified.
Table 6, I think that introducing the acronymns F-PID and F-SLC very early on in the paper would be helpful; that two different FLC’s are considered, an F-PID and an F-SLC.
Figure at Line 287. The figure is mis-numbered, it is 10 not 1. All subsequent figures are mis-numbered, too.
Line 337, Table 8, Is “maximum overshoot” of F_PID really correct? FOS doesn’t seemed to have equilibrium depth in relevant figure.
Reviewer 2 Report
The work is very interesting.
It seems to me that not all the generalised system coordinates from Figure 3 are described, explained.
Relationships (3), (4) and (5) contain numbers without units.
In (8) the variable v is a vector and should be marked in bold, straight font.
The individual components (8) should be described in more detail.
It is not shown how depth affects the equation (10).
There is no diagram of a device containing sensors, actuators, computer, power supply, etc. at the work.The question arises whether the dynamics of sensors and actuators are taken into account in the simulation.
The method of selection of PID regulator settings was also not presented. Optimization of the PID regulator settings may sufficiently improve its operation.
